# Membrane Localization and Phosphorylation of Indoleamine 2,3-Dioxygenase 2 (IDO2) in A549 Human Lung Adenocarcinoma Cells: First Steps in Exploring Its Signaling Function

**DOI:** 10.3390/ijms242216236

**Published:** 2023-11-12

**Authors:** Chiara Suvieri, Francesca De Marchis, Martina Mandarano, Sara Ambrosino, Sofia Rossini, Giada Mondanelli, Marco Gargaro, Eleonora Panfili, Ciriana Orabona, Maria Teresa Pallotta, Maria Laura Belladonna, Claudia Volpi

**Affiliations:** 1Section of Pharmacology, Department of Medicine and Surgery, University of Perugia, 06129 Perugia, Italy; chiara.suvieri@unipg.it (C.S.); sara.ambrosino@studenti.unipg.it (S.A.); sofia.rossini@unipg.it (S.R.); giada.mondanelli@unipg.it (G.M.); marco.gargaro@unipg.it (M.G.); eleonora.panfili@unipg.it (E.P.); ciriana.orabona@unipg.it (C.O.); maria.pallotta@unipg.it (M.T.P.); marialaura.belladonna@unipg.it (M.L.B.); 2Institute of Biosciences and Bioresources, Research Division of Perugia, National Research Council (CNR), 06128 Perugia, Italy; francesca.demarchis@ibbr.cnr.it; 3Section of Anatomic Pathology and Histology, Department of Medicine and Surgery, University of Perugia, 06129 Perugia, Italy; martina.mandarano@unipg.it

**Keywords:** IDO2, lung adenocarcinoma, A549, membrane localization, tyrosine phosphorylation

## Abstract

Indoleamine 2,3-dioxygenase 2 (IDO2) is a paralog of Indoleamine 2,3-dioxygenase 1 (IDO1), a tryptophan-degrading enzyme producing immunomodulatory molecules. However, the two proteins are unlikely to carry out the same functions. IDO2 shows little or no tryptophan catabolic activity and exerts contrasting immunomodulatory roles in a context-dependent manner in cancer and autoimmune diseases. The recently described potential non-enzymatic activity of IDO2 has suggested its possible involvement in alternative pathways, resulting in either pro- or anti-inflammatory effects in different models. In a previous study on non-small cell lung cancer (NSCLC) tissues, we found that IDO2 expression revealed at the plasma membrane level of tumor cells was significantly associated with poor prognosis. In this study, the A549 human cell line, basally expressing IDO2, was used as an in vitro model of human lung adenocarcinoma to gain more insights into a possible alternative function of IDO2 different from the catalytic one. In these cells, immunocytochemistry and isopycnic sucrose gradient analyses confirmed the IDO2 protein localization in the cell membrane compartment, and the immunoprecipitation of tyrosine-phosphorylated proteins revealed that kinase activities can target IDO2. The different localization from the cytosolic one and the phosphorylation state are the first indications for the signaling function of IDO2, suggesting that the IDO2 non-enzymatic role in cancer cells is worthy of deeper understanding.

## 1. Introduction

The immunoregulatory metabolite kynurenine (Kyn) can be produced via tryptophan (Trp) degradation operated in the heme-containing enzymes tryptophan 2,3-dioxygenase 2 (TDO2) and indoleamine 2,3-dioxygenase 1 (IDO1) [1]. TDO2, selectively expressed in the liver, is the primary enzyme responsible for the metabolism of dietary tryptophan and controls the homeostasis of this essential amino acid. IDO1, acting on a wide array of indoleamine-containing substrates, has a high affinity for Trp and an enzymatic biological function based on the immunosuppressive effect caused by Trp depletion and Kyn accumulation, inhibiting effector T cell responses and promoting Treg and dendritic cell (DC) tolerance [2,3,4]. Another enzyme that potentially converts Trp into Kyn is the IDO1 paralog indoleamine 2,3-dioxygenase 2 (IDO2) discovered by Ball and coworkers in 2007 [5].

IDO2, mapped to chromosome 8 in humans and mice, is encoded by a gene located adjacent to and downstream of *IDO1*. The two genes were generated from a more ancient proto-*IDO* gene duplication that occurred before the divergence of marsupial and eutherian (placental) mammals [5,6,7]. IDO1 is widely expressed in different cell types, like the endothelial cells of the placenta and lung or lymphoid tissues. On the contrary, *IDO2* mRNA expression is confined to the liver, cerebral cortex, and kidney. Multiple pro-inflammatory stimuli have been proposed as IDO2 regulators, though with relatively less strength than IDO1 [8,9,10,11]. Despite displaying high sequence homology (43%), IDO1 and IDO2 have different affinities for Trp. IDO1 has a high rate of Trp catalysis (K*_m_* around 7–22 μM), while IDO2 has been reported to have a very low catalytic efficiency with a high K*_m_* value (6.8–9.4 mM), 100-fold higher than the physiological l-tryptophan concentrations [12,13]. Based on this evidence, several possible hypotheses have been made for IDO2 activity and function, including the erroneous K*_m_* evaluation due to the interference of reducing reagents commonly used to dose indoleamine 2,3-dioxygenase activity [14], the existence of a natural substrate different from Trp, and a functional role distinct from Trp catabolism [15,16]. In support of the latter hypothesis, researchers have found that *Ido2* deletion does not decrease Kyn systemic levels in knockout mice while affecting IDO1-dependent T cell suppression and Treg induction [17], suggesting an *Ido1*–*Ido2* genetic interaction and a possible functional role of *Ido2* in the modulation of immune responses. However, this latter aspect remains ambiguous. Moreover, an untargeted analysis of metabolites produced by cells overexpressing human or murine IDO2 revealed that no amino acid nor any other compound commonly found in the cell line culture medium is specifically metabolized by IDO2 (data obtained in our laboratory and not shown), confirming that the similarity between IDO1 and IDO2 in the amino acid sequence does not necessarily result in the same metabolic activity. For IDO2, a pro-inflammatory function has been described in B cells in the initiation, progression, and severity of autoimmune arthritis in both the in vivo model of KRN.g7 mice, which is genetically deficient of the *Ido2* gene, and the specific silencing of *Ido2* [18,19]. Nevertheless, in other studies, *Ido2* appears to have an anti-inflammatory role in a psoriasis-like inflammation model since its deletion exacerbates disease symptoms and increases the number of IL-17-positive lymphocytes infiltrating the dermis [20].

The expression of IDO2 is upregulated in various cancers [21], including non-small cell lung cancer (NSCLC) [22], pancreatic [23,24], colon, gastric, and renal tumors [25,26], and medullary thyroid carcinoma [27]. Liu et al. assessed the biological value of IDO2 in mouse B16/BL6 melanoma cells, which showed a significantly reduced proliferation with a cell cycle arrest in the G1 phase, high apoptosis rate, and reduced cell migration when the constitutively highly expressed *Ido2* gene was silenced. In the same study, the in vivo onset of tumor growth was delayed in mice injected with *Ido2*-silenced cells [28]. IDO2 overexpression is recurrent in pancreatic ductal adenocarcinoma (PDAC) tumors and is involved in tumorigenesis mechanisms, as demonstrated by the improved disease-free survival recorded in adjuvant-radiotherapy-treated PDAC patients featuring single nucleotide polymorphisms (SNPs) thought to completely inactivate the already negligible Trp-catalytic activity of IDO2 [23,29].

In previous studies, we investigated the involvement of IDO2 SNPs in a large cohort of NSCLC patients and healthy matched controls. When evaluating cancer risk, we found a highly significant incidence of the R248W genotype in NSCLC patients compared to the control group [30]. Moreover, strong evidence of a significant correlation between IDO2 expression and poor NSCLC prognosis was revealed in a recent study performed through an extensive immunohistochemical analysis of NSCLC specimens and having detected IDO2 presence in tumor cells but not in normal lung tissues [22]. Among the analyzed histotypes, adenocarcinoma showed the highest IDO2 expression associated with high intratumoral/mixed tumor-infiltrating lymphocyte localization. In the same study, 83% of tumors showed a membrane reinforcement staining of IDO2 that, in 51% of the cases, localized at the basolateral side of the cell membrane between tumor and stromal tissue [22].

Based on these previous data revealing that IDO2 correlates with a poor prognosis and can be stained at the basolateral side of tumor—but not normal—cell membrane in NSCLC, we asked whether this IDO2 surface localization could be instrumental for its own still not elucidated alternative function (i.e., different from the catalytic one) in cancer cells. Upon evaluating the possible functions of cell membrane proteins, a putative signaling function contributing to the neoplastic cell program seemed to us to be consistent with both the differential IDO2 expression observed in tumor versus normal cells and the already known moonlighting activity of the IDO2’s paralog IDO1.

Thus, in this present work, we display the very first results of an ongoing study aimed to verify the hypothesis of a signaling function for IDO2 in lung tumor cells. For this purpose, we chose the human lung adenocarcinoma cell line A549, which basally expresses IDO2 and is widely used as a lung carcinoma/infectious model and for drug discovery. The in vitro experiments point to investigating the localization of IDO2 protein and its possible interplay with molecular partners to better explain the results observed in NSCLC specimens.

## 2. Results

### 2.1. A549 Cells Basally Express IDO2

In previous studies, we found that the IDO2 protein was detectable via immunostaining in NSCLC specimens, most of which presented staining reinforcement at the membrane level [22]. We used the lung adenocarcinoma human cell line A549 as an in vitro cell model and the A549/hIDO2 transfectant as a positive control overexpressing the human *IDO2* gene for more insights into the cellular localization of IDO2 in lung tumor cells. First, IDO2 expression was verified at the transcript and protein levels. After 40 cycles of end-point RT-PCR, the IDO2 product amplified from A549 cDNA was of the same size, albeit less abundant than the A549/hIDO2 positive control. The human actin beta (*ACTB*) was used as a housekeeping gene for sample normalization (Figure 1A). In the immunoblot analysis, the IDO2 protein was found to be basally expressed in this primary lung tumor-derived cell line and gave a signal at the same molecular weight detected for the IDO2 protein overexpressed in the A549/hIDO2 transfectant. The immunoblot with anti-β-tubulin was used as the loading control (Figure 1B). To investigate whether IDO2 endogenously expressed in A549 cells was capable of degrading Trp, Kyn production was measured in the culture supernatant of either A549 cells or A549/hIDO2 transfectant. In line with the very low Trp catalytic activity already reported for IDO2 [12,13], Kyn concentrations in the supernatants of both the A549 cell line and A549/hIDO2 transfectant were comparable to that recorded in the medium incubated without cells (Figure 1C). To confirm the anti-hIDO2 antibody’s ability to recognize IDO2 protein, *IDO2* siRNA-mediated silencing was used in A549 cells. After 48 h of treatment, the IDO2 protein level was downregulated in comparison to the negative control (Figure 1D). Thus, because the IDO2 protein is detectable at its basal expression level and not active in catabolizing Trp into Kyn in these cells, the A549 human lung carcinoma cells were chosen as a suitable model to explore IDO2 cellular localization and possible alternative functions different from the enzymatic one.

### 2.2. IDO2 Is Localized at the Membrane Level in A549 Cells

The cell localization of IDO2 in A549 cells was investigated by immunostaining paraffin-embedded cells. The IDO2 staining, absent in the control samples incubated without the primary antibody (Figure 2A, upper and lower left panels), was visible at the plasma membrane level in the A549 cells (Figure 2A, upper right panel), agreeing with what was reported in NSCLC patients’ tissues [20], and appeared much stronger in the A549/hIDO2 transfectant overexpressing IDO2 (Figure 2A, lower right panel). IDO2 plasma membrane localization was further assessed in A549 cells by immunofluorescence analysis (Figure 2B). On the basis of these results, we hypothesize that IDO2 is prevalently located on the plasma membrane, although we cannot rule out its possible localization in other membranes, such as the Golgi and ER membranes.

Since, in Western blotting experiments, the anti-hIDO2 antibody non-selectively recognized other proteins besides IDO2 (Appendix A), we used a sucrose isopycnic gradient analysis to fractionate subcellular organelles from A549 cells and gain more insights into the specific cell compartment where IDO2 was localized. The immunoblot via the anti-hIDO2 antibody on the obtained different-density cell fractions revealed that IDO2 was completely absent in low-density (#1–3) and high-density (#14–17) cytosol fractions. In contrast, IDO2 was detectable in medium-density fractions (#9–13), with a very strong signal in fraction #10 (Figure 2C). The re-immunoblotting with an antibody recognizing E-cadherin, a specific epithelial cell adhesion molecule mainly located on the plasma membrane, indicated that IDO2^+^ and E-cadherin^+^ fractions overlapped, both showing stronger signals at fraction #10, thus suggesting a likely cellular colocalization of the two molecules. Therefore, these data indicated that IDO2 is not confined to the cytosol in A549 cells but rather localized at the membrane level, a peculiar position suggesting its potential role as a signaling molecule.

### 2.3. IDO2 Is Tyrosine Phosphorylated in A549 Cells

In immunoprecipitation experiments, the tyrosine phosphorylation of IDO2 has been investigated since its polypeptide contains tyrosine phosphorylation consensus sequences that might be possible targets of kinase activities, triggering the putative signaling function of membrane-localized IDO2. As a result, a whole cell lysate was prepared from A549, and an aliquot of it was saved to be assayed as a pre-immunoprecipitation protein content control via anti-hIDO2 immunoblotting (Figure 3A, WCL right panel). After immunoprecipitation, either in the presence or absence of an anti-phospho-Tyr antibody, tyrosine-phosphorylated proteins were immunoblotted with the anti-hIDO2 antibody, revealing the presence of a tyrosine-phosphorylated form of IDO2 in A549 cells (Figure 3A, IP left panel). The selectivity of the anti-phospho-Tyr antibody was demonstrated by dephosphorylation experiments with alkaline phosphatase. The treatment of A549 whole cell lysate with alkaline phosphatase (FastAP) was able to efficiently reduce protein phosphorylation, while, in the presence of phosphatases inhibitors, an extensive phosphorylation pattern was evident (Figure 3B). Moreover, FastAP-pre-treated or untreated A549 whole cell lysates were immunoprecipitated with the anti-phospho-Tyr and immunoblotted with the anti-hIDO2 antibody. A signal corresponding to the tyrosine-phosphorylated form of IDO2 was revealed only in the FastAP-not-treated sample (Figure 3C).

Overall, these data demonstrated that IDO2 could exist in a tyrosine-phosphorylated state in A549 cells. Indeed, human IDO2 possesses two ITIM domains like IDO1: while the so-called ITIM2 shows a high homology sequence and a similar position in both the IDO proteins (VYEGF in human IDO1 and MYEGV in human IDO2), the putative IDO2’s ITIM1 motif (IFYAGI) differs from that of IDO1 (VPYCQL). Moreover, the YENM motif—useful for the binding of IDO1 to the phosphoinositide 3-kinase (PI3K) [31]—is absent in IDO2 [30] (Figure 3D). The findings of IDO2 membrane localization and tyrosine phosphorylation support the hypothesis of a prevalent—if not unique—signaling rather than the metabolic activity of IDO2 in A549 cells.

## 3. Discussion

Since its discovery in 2007 [5], many efforts have been made to understand the function of IDO2. Although it was evident from the first studies performed that IDO2 was endowed with a negligible Trp-metabolizing activity compared to IDO1 and TDO2, it has been counted as one of the three enzymes involved in Trp catabolism for many years [32]. In addition to this assumption, the high amino acid homology between IDO1 and IDO2, either human or murine, and the absence of a crystal structure of IDO2, prompted investigators to hypothesize an enzymatic immunoregulatory role for IDO2, which is functionally related to its paralog IDO1. Nowadays, though there is much evidence connecting IDO1 and IDO2, similar functions for the two proteins have not always been determined. IDO1 and IDO2 appear to play opposite roles in B-cell-mediated immunity, with IDO1 inhibiting and IDO2 driving the inflammatory responses of B cells [33]. In particular, in the KRN.g7 mouse model of autoimmune arthritis, IDO1 and IDO2 appear to have contrasting roles in regulating B-cell-mediated immune responses. Using *Ido1* and *Ido2* single and double knockout mice clarified that IDO1 is responsible for T cell suppression, while IDO2, directly acting in B cells, mediates the proinflammatory response of autoimmune processes in an IDO1-independent manner [19]. Moreover, while IDO1 expression is described to have a protective value in multiple sclerosis animal models, IDO2 deletion does not seem to affect the course of the pathology in experimental autoimmune encephalomyelitis [34]. In contrast, in a psoriasis-like dermatitis model, *Ido2*, but not *Ido1*, knockout mice were associated with exacerbated symptoms, with higher CD4^+^ and CD8^+^ T cell infiltration and IL-17-positive cells [20]. In addition, *Ido2* deletion was associated with higher mortality in the LPS-induced endotoxin shock model and correlated with the increased production of inflammatory cytokines (including IL-6) in serum [35].

Besides its role in murine models of autoimmune and inflammatory diseases, IDO2 appears to have a prominent role in the onset, development, and prognosis of many types of human cancers [25,30], in striking analogy with IDO1 [16,36,37]. The upregulation of IDO1 in several cancer cell types is well demonstrated, acting as a mediator of tumor immune escape and, therefore, is often associated with a worse outcome [8,38]. As a result, multiple compounds targeting IDO1 catalytic activity have been developed and considered possible drug candidates in cancer immunotherapy [38,39] but with no evident beneficial effects so far [40]. The failure of IDO1 catalytic inhibitors can be explained by the observation that the signaling activity of IDO1 is retained either in dendritic cells or in tumor cells [41], despite the selective loss of the IDO1’s Trp-metabolizing function. Nowadays, it is widely accepted that IDO1 protein is not only endowed with its well-defined catalytic activity but can also act as a non-enzymatic signaling platform via its phosphorylation domains, i.e., the ITIM1 and ITIM2 motifs, and the YENM sequence [31]. Interestingly, while the Trp-degrading IDO1 enzyme is mostly located in the cytosol, the signaling IDO1 protein is localized to the early endosomes, which are active subcellular signaling locations [31]. Although the precise subcellular localization of human IDO2 is still to be defined and the function of its ITIM motifs is still unknown, it could be speculated that the IDO2’s proximity to the plasma membrane is functional to a signaling activity and ITIMs act as connection points to recruit adaptor proteins containing SH2 domains, as it occurs in IDO1. Indeed, it has been previously demonstrated that SH2-containing tyrosine phosphatases (SHPs) and SOCS3 can bind IDO1’s phosphorylated ITIM1 and ITIM2, respectively [41,42], and that Src kinase is involved in activating IDO1 signaling in DCs [43]. If analog mechanisms also operate in the IDO2 molecule, this is something that should be thoroughly investigated.

Our previous findings, reported via an immunohistochemical analysis of NSCLC specimens, unveiled a membrane reinforcement of IDO2 staining in most analyzed tumors, half of which presented higher IDO2 expressions (i.e., stain reinforcement) on the basolateral side of the tumor cell plasma membrane at the interface between tumor and stromal tissue and without an apical immunolabel [22]. These findings prompted us to deepen our knowledge about the cell compartment where IDO2 localizes and unveils a possible signaling pathway initiated via the association of IDO2 with putative molecular partners. The A549 human lung adenocarcinoma cell line, a well-accepted in vitro lung carcinoma model, allowed us to reproduce some observations reported in our previous immunohistochemical study on surgical specimens from NSCLC patients and find additional evidence of IDO2 phosphorylation, suggesting it might be involved in signaling pathways. IDO2 protein, basally detectable via end-point RT-PCR and immunoblot experiments, displayed negligible tryptophan activity in A549 cells; in this way, Kyn concentration was similar to that recorded in the medium incubated without cells. The plasma membrane localization of IDO2, already observed in surgical specimens from NSCLC patients, was confirmed by immunostaining both paraffin-embedded A549 cells and A549/hIDO2 transfectants. For the first time, a sucrose isopycnic gradient analysis documented such findings, proving the pick overlap of both the IDO2 and plasma membrane protein E-cadherin at fraction #10, thus suggesting that IDO2 could be a peripheral or integral membrane protein. The never previously described evidence of IDO2 tyrosine phosphorylation in unstimulated A549 cells corroborated our hypothesis that IDO2 membrane localization is functional for its signaling activity.

It could be speculated that both the membrane localization and phosphorylation state of IDO2 found in A549 cells are functional for signaling activity, possibly related to a neoplastic cell phenotype. As a matter of fact, this hypothesis can be corroborated by the evidence of IDO2 presence in several human tumors and from the lack of a well-defined IDO2 function alternative to the enzymatic one [25,30]. These findings, besides the documented staining of IDO2 in cancer but not in normal NSCLC cells [22], suggest a potential relationship between the expression/signaling of IDO2 and a possible control loss in the common mechanisms of cell cycle regulation and proliferation. Moreover, our study on IDO2 polymorphism supports the hypothesis of an IDO2 biological function alternative to the catalytic one. In the human IDO2 sequence, a high prevalence of two SNPs has been described. The first is rs10109853, corresponding to a substitution of arginine to tryptophan in position 248 of the human amino acid sequence (R248W), which leads to a further >90% reduction in its catalytic activity. The second is rs4503083, a nonsense mutation that generates a premature stop codon (Y359X) [7]. The absence of enzymatic activity because of the presence of one or both above-mentioned SNPs has been elegantly demonstrated [17,29]. Overall, these observations suggest that, in tumor cells, rather than a Trp degrading activity, IDO2 possesses a still unexplored non-enzymatic function taking place at the cell membrane level, involving its phosphorylation and being related to the activation of a neoplastic cell program. Although preliminary and limited, the data from the present study pave the way for our ongoing investigation with *IDO2* knockout A549 cells and aim to deepen the study of putative IDO2 signaling in lung tumor cells.

## 4. Materials and Methods

### 4.1. Cell Culture and End-Point RT-PCR

The A549 human lung adenocarcinoma cell line, obtained from the American Type Culture Collection (ATCC, Manassas, VA, USA), was cultured according to standard procedures in RPMI-1640 medium, supplemented with a 10% heat-inactivated fetal bovine serum (FBS) and 2 mM of l-glutamine and antibiotics (100 U/mL penicillin, 100 μg/mL streptomycin). A lipofectamine 3000 reagent (Thermo Scientific, Waltham, MA, USA) and a pEF-BOS-based vector containing *IDO2* wild-type gene, generated as previously described [41], were used to transfect A549 cells according to the manufacturer’s instructions. After puromycin selection (0.5 μg/mL), A549/hIDO2 transfectant was characterized for IDO2 protein expression. Cells were incubated at 37 °C in a 5% CO_2_ and humidified atmosphere. In this present study, the A549/hIDO2 transfectant was used as an IDO2-overexpressing control of the untransfected A549 cell line, as shown in Figure 1 and Figure 2.

For the end-point RT-PCR, RNA was extracted with TRIzol Reagent (Thermo Scientific, Waltham, MA, USA) and retro-transcribed with a QuantiTect^®^ Reverse Transcription kit (QIAGEN, Hilden, Germany). cDNA was amplified using Platinum Taq DNA polymerase (Thermo Scientific, Waltham, MA, USA) for 40 cycles for *IDO2* or 30 cycles for the human actin beta (*ACTB*) housekeeping gene used for sample normalization. Primers: *IDO2* FORW 5′- TGG GAT AAA GGC TCT TGT TC-3′; *IDO2* REV 5′-TGG TGA TGT CCT GAA TAG AC-3′; *ACTB* FORW 5′-CTC GTC GTC GAC AAC GGC T-3′; *ACTB* REV 5′- TCA GGG TGA GGA TGC CTC TC-3′.

### 4.2. Kyn Determination

The enzymatic activity that catabolized Trp into Kyn was measured in vitro in the culture supernatants of A549 cells and A549/hIDO2 transfectant (2 × 10^5^ cells/mL per well in a 12-well plate) after 48 h of incubation. Kyn concentration was detected using a Perkin Elmer (Waltham, MA, USA), series 200, HPLC instrument, combined with a Kinetex^®^ C18 column (250 × 4.6 mm, 5 μm, 100 A; Phenomenex, Torrance, CA, USA), maintained at the temperature of 25 °C and pressure of 1800 PSI. A mobile phase containing 10 mM NaH_2_PO_4_ pH 3.0 (99%) and methanol (1%) (Sigma-Aldrich, Merk, Darmstadt, Germany), with a flow rate of 1 mL/min, was used, and a UV detector identified Kyn and Trp at 360 nm and 220 nm, respectively. The software TotalChrom v. 6.3.1 was used for evaluating the concentrations of Kyn in samples using a calibration curve. The detection limit of the analysis for Kyn was 0.05 μM.

### 4.3. Western Blotting

In the immunoblot experiment, the A549 whole cell lysate was prepared from 2 × 10^5^ cells and analyzed using SDS-PAGE. As a positive control, the whole cell lysate from A549/hIDO2 transfectant, overexpressing the human *IDO2* gene, was also included. IDO2 protein expression was assessed via Western blotting with the rabbit polyclonal anti-hIDO2 antibody I23O2 (Aviva Systems Biology, San Diego, CA, USA) combined with an appropriate horseradish peroxidase-conjugated antibody (Merk Millipore, Burlington, MA, USA), followed by enhanced chemiluminescence (ECL, Bio-Rad, Hercules, CA, USA). As a loading control, β-tubulin was revealed with a specific mouse monoclonal antibody (clone AA2, Merk Sigma-Aldrich, Darmstadt, Germany).

### 4.4. siRNA Transfection

For human IDO2 silencing, 5 × 10^4^ A549 cells were plated in a 6-well plate overnight at 37 °C. The day after, cells were transfected by Lipofectamine™ 3000 (Thermo Scientific, Waltham, MA, USA), according to the manufacturer’s instructions, with 100 pmol of Silencer™ Select human IDO2 siRNA (ID s46747, Thermo Scientific, Waltham, MA, USA) or Silencer™ Select Negative Control No.1 (Thermo Scientific, Waltham, MA, USA) and incubated for 24 or 48 h. Then, cells were recovered and IDO2 protein expression was assessed by means of Western blotting analysis using the rabbit polyclonal anti-hIDO2 antibody I2302 (Aviva Systems Biology, San Diego, CA, USA). Anti–β-tubulin (clone AA2, Merk Sigma-Aldrich, Darmstadt, Germany) immunoblot was used as a loading control.

### 4.5. Immunoprecipitation and De-Phosphorylation Assay

For the immunoprecipitation (IP) of phosphorylated IDO2, 12 × 10^6^ A549 cells were lysed for 30 min on ice in 400 µL of lysis buffer (20 mM Tris-HCl pH 7.4, 50 mM NaCl, 1% Triton-100) containing cOmplete™ Protease Inhibitor Cocktail (Roche, Merk, Darmstadt, Germany) and phosphatase inhibitors (Thermo Scientific, Waltham, MA, USA). After centrifugation, the supernatant was incubated overnight at 4 °C under rotary agitation with phospho-tyrosine (P-Tyr-1000) MultiMab™ Rabbit mAb mix (Cell Signaling, Danvers, MA, USA) (2 µL of Ab in 400 µL of lysate/sample) or in the absence of it in the negative control sample. Before antibody incubation, 20 µL of WCL was taken from each sample for subsequent Western blotting analysis. After adding the Protein-A Sepharose (PAS) beads (Sigma-Aldrich, Merk, Darmstadt, Germany) (30 µL/sample), the samples were incubated for 1 h at 4 °C under rotary agitation. The negative control was incubated only with PAS beads without previous incubation with the P-Tyr-1000 mAb mix. To eliminate possible non-specific interactions, IP samples were centrifuged at 800 rpm for 5 min at 4 °C and washed 3 times with a washing buffer (20 mM Tris-HCl pH 7.4, 150 mM NaCl, 0.5% Triton-100). IDO2 immunoblot detection was performed with the rabbit polyclonal anti-hIDO2 antibody I23O2 (Aviva Systems Biology, San Diego, CA, USA), as described above in the Western blotting analysis.

For IP after protein de-phosphorylation, the WCL was prepared from 20 × 10^6^ untreated A549 cells as previously described and split into two samples to be treated with FastAP Thermosensitive Alkaline Phosphatase (3 µL each 200 µg of proteins) (Thermo Scientific, Waltham, MA, USA) or phosphatase inhibitors, for 1 h at 37 °C. Then, 10 µL was taken from both samples at the conclusion of the incubation and used on Western blotting analysis with anti-phospho-tyrosine P-Tyr-1000 MultiMab™ Rabbit mAb mix and anti-β-actin (clone AC-40, Merk Sigma-Aldrich, Darmstadt, Germany) as the loading control.

Then, samples were incubated as previously described with phospho-tyrosine (P-Tyr-1000) MultiMab™ Rabbit mAb mix (Cell Signaling, Danvers, MA, USA). Phosphorylated proteins were immunoprecipitated as a standard procedure with PAS beads. Washed immunoprecipitated samples were run on SDS-PAGE and analyzed by Western blotting. Tyrosine-phosphorylated IDO2 detection was achieved using the specific rabbit polyclonal anti-hIDO2 antibody I23O2 (Aviva Systems Biology, San Diego, CA, USA).

### 4.6. Immunocytochemical Analysis

A549 and A549/hIDO2 cells (5 × 10^6^ each) were collected and fixed in 5 mL of 4% formaldehyde (10% formalin, neutral buffered) (Sigma, Merk, Darmstadt, Germany). Subsequently, a cell block was obtained via the Hologic protocol relating to the Cellient™ Automated Cell Block System (Hologic, Marlborough, MA, USA): the formalin-fixed cell cultures were at first centrifuged at 1727 rpm for 10 min. After supernatant removal, the cell precipitate was transferred into a vial of Preservcyt™ solution (buffered methanol-based solution) of the ThinPrep™ system (Hologic, Marlborough, MA, USA) and processed via the Cellient™ instrument for 45 min, concentrating the cells and distributing them in a thin layer into a paraffin cell block. Subsequently, 4 µm sections were prepared and placed on positively charged slides for immunocytochemical staining via the BOND III fully automated immunohistochemistry stainer (Leica Biosystems, Nußloch, Germany). Heat-induced antigen retrieval was obtained by incubating slides for 20 min with the ready-to-use citrate-based pH 6.0 Bond™ Epitope Retrieval Solution 1 (Leica Biosystems, Newcastle Upon Tyne, UK). Subsequent incubations were performed with the polyclonal anti-hIDO2 primary antibody I23O2 for 15 min (dilution 1: 500, Aviva Systems Biology, San Diego, CA, USA) and the ready-to-use Bond™ Polymer Refine Detection System (Leica Biosystems, Newcastle Upon Tyne, UK). This reagent includes a peroxide block, a secondary antibody (rabbit anti-mouse IgG), an anti-rabbit Poly–HRP-IgG to localize the post-primary rabbit antibodies, chromogen 3,3′-diaminobenzidine tetrahydrochloride hydrate (DAB) substrate to visualize the complex via a brown precipitate, and hematoxylin for nuclear counterstaining. An appropriate negative control was included during the test using the same immunostaining protocol but without the primary antibody incubation.

### 4.7. Immunofluorescence Analysis

A549 cells (5 × 10^5^) were plated over polylysine-coated coverslips in a 6-well plate for 24 h at 37 °C. Cells were fixed for 20 min with 4% formaldehyde (10% formalin, neutral buffered) (Sigma, Merk, Darmstadt, Germany) in PBS and non-specific antibody binding sites were blocked for 1 h at room temperature with 1% BSA in PBS (blocking buffer) and 5% goat serum (Thermo Scientific, Waltham, MA, USA). Cells were stained overnight at 4 °C with anti-hIDO2 antibody I23O2 (Aviva Systems Biology, San Diego, CA, USA). The negative control was incubated without the primary antibody. Samples were first stained with the secondary biotinylated goat anti-rabbit IgG antibody (Vector Laboratories Fisher Scientific, Waltham, MA, USA) at 1:500 for 1 h at room temperature and then with streptavidin-Alexa Fluor 594 (Jackson Immunoresearch, Bar Harbor, ME, USA) at 1:1000 for 30 min, all in blocking buffer. Nuclei were counterstained with Hoechst 33342 (Thermo Scientific, Waltham, MA, USA) at 1:500 for 10 min at room temperature. Slides were mounted with ProLong™ Gold Antifade Mountant (Thermo Scientific, Waltham, MA, USA). Images were captured by a Nikon inverted microscope, equipped with a spinning disk, and analyzed by ImageJ software v1.54f.

### 4.8. Isopycnic Gradient Analysis

For subcellular fractionation, isopycnic sucrose gradient centrifugation was performed as previously described [31]. Briefly, 5 × 10^6^ A549 cells were resuspended in 1 mL of sucrose buffer (12% sucrose, 10 mM KCl, 2 mM MgCl_2_, and 100 mM Tris-HCl pH 7.2) and homogenized. Then, the cell homogenate was loaded on top of a continuous sucrose gradient (16–55% *w*/*w*), and after ultracentrifugation at 141,000× *g* for 4 h at 4 °C, seventeen fractions were recovered. From each fraction, proteins were precipitated via TCA and resuspended in 50 µL of 1× loading dye. Samples were analyzed via SDS-PAGE and immunoblotting for the presence of IDO2 (I23O2 rabbit polyclonal antibody, Aviva Systems Biology, San Diego, CA, USA) and E-cadherin (rabbit mAb clone SP64, Abcam, Cambridge, UK) with the appropriate horseradish peroxidase-conjugated antibody (Merk Millipore, Burlington, MA, USA), followed by enhanced chemiluminescence (ECL, Bio-Rad, Hercules, CA, USA).

### 4.9. Statistical Analysis

For statistical analysis, the GraphPad Prism 8.0.1 software for Windows (GraphPad) was used. Data are expressed as the mean ± S.D., and statistical significance was determined via one-way ANOVA analysis. A *p*-value ≤ 0.05 was considered statistically significant.

## 5. Conclusions

In conclusion, taking the very first steps to demonstrate our hypothesis of a putative signaling role for IDO2 in tumor cells, we found that the human lung adenocarcinoma cell line A549 is a suitable in vitro model to study the cellular localization of IDO2 and its still-not-completely elucidated non-enzymatic function in tumor cells. The evidence that IDO2 protein, basally expressed in A549 cells, localizes at the plasma membrane level and is tyrosine-phosphorylated supports the possible hypothesized role as a signaling molecule in tumor cells for this protein. Further signaling analyses (e.g., on kinases targeting IDO2 and molecular adaptors) and functional studies (e.g., on the proliferative and metastatic potential of IDO2-expressing cancer cells) are worth conducting to dissect IDO2’s putative role in tumor cell programs, thus aiming to define IDO2 as a suitable drug target in cancer therapy.

## Figures and Tables

**Figure 1 ijms-24-16236-f001:**
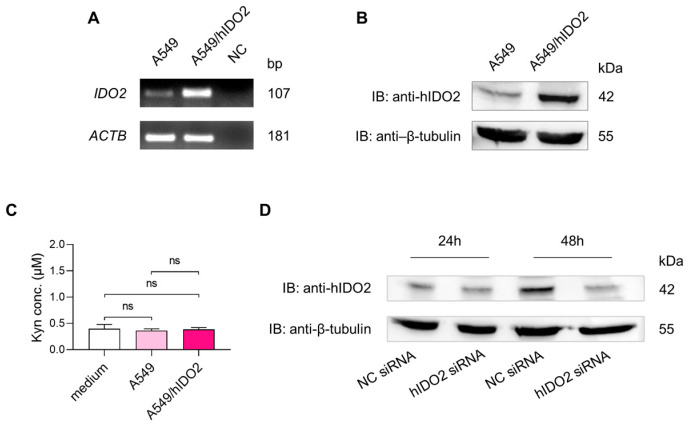
IDO2 transcript and protein are basally detectable in A549 cells and associated with negligible Kyn production. *IDO2* expression and catalytic activity were investigated in the A549 cell line and A549/hIDO2 transfectant, which was used as a positive control overexpressing the human *IDO2* gene. (**A**) *IDO2* expression was analyzed using end-point RT-PCR (40 cycles). The human actin beta (*ACTB*) was the housekeeping gene for sample normalization (30 cycles). NC, negative control (i.e., without cDNA). One experiment representative of three. (**B**) In A549 and A549/hIDO2 whole cell lysates, IDO2 protein expression was assessed via Western blotting with the rabbit polyclonal anti-hIDO2 antibody I23O2. Anti-β-tubulin immunoblot served as a loading control. One experiment representative of three. (**C**) The catalytic-activity-degrading Trp was measured using HPLC as Kyn released in the culture supernatant of either A549 cells or A549/hIDO2 transfectant. Data (mean ± SD) are the results of three independent measurements. The culture medium incubated without cells is also shown. One-way ANOVA was used for the analysis. ns, not significant. (**D**) IDO2 protein expression in A549 cells was assessed after treatment for 24 and 48 h with 100 pmol of hIDO2 siRNA, or negative control (NC) siRNA. The rabbit polyclonal anti-hIDO2 antibody I23O2 was used to detect the IDO2 protein, while β-tubulin was analyzed for protein loading.

**Figure 2 ijms-24-16236-f002:**
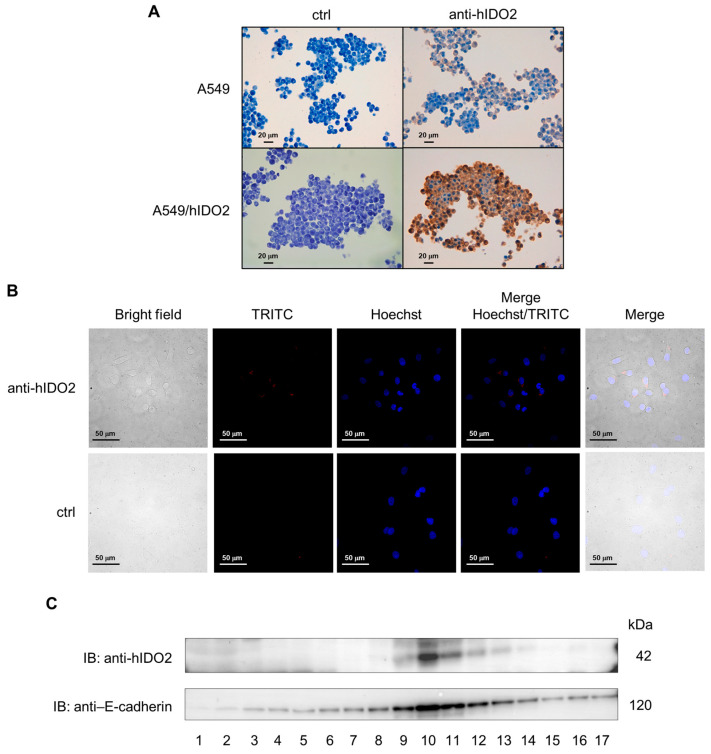
Membrane localization of IDO2 protein in A549 cells. (**A**) Immunocytochemical analysis of IDO2 in paraffin-embedded A549 cells and A549/hIDO2 transfectant. Representative images of membranous staining in sections incubated in the presence of I23O2 anti-hIDO2 antibody (upper and lower right panels) or the absence of it (ctrl, negative control; upper and lower left panels). A549/hIDO2 transfectant was included as a positive control overexpressing the human *IDO2* gene. Original magnification 400×; scale bar: 20 μm. (**B**) Images of IDO2 immunofluorescence in A549 cells stained in the presence of anti-hIDO2 antibody (in red, TRITC) or with secondary alone as the control (ctrl, biotinylated goat anti-rabbit IgG antibody combined with streptavidin-Alexa Fluor 594). Original magnification 60×, scale bar: 50 µm. (**C**) Analysis of IDO2 cellular localization via isopycnic sucrose gradient of A549 fractions prepared from whole cell lysate. Fractions were immunoblotted with anti-hIDO2 I23O2 and anti-E-cadherin as a marker of the plasma membrane compartment. One experiment is representative of three (anti-hIDO2 IB) or one (anti–E-cadherin IB).

**Figure 3 ijms-24-16236-f003:**
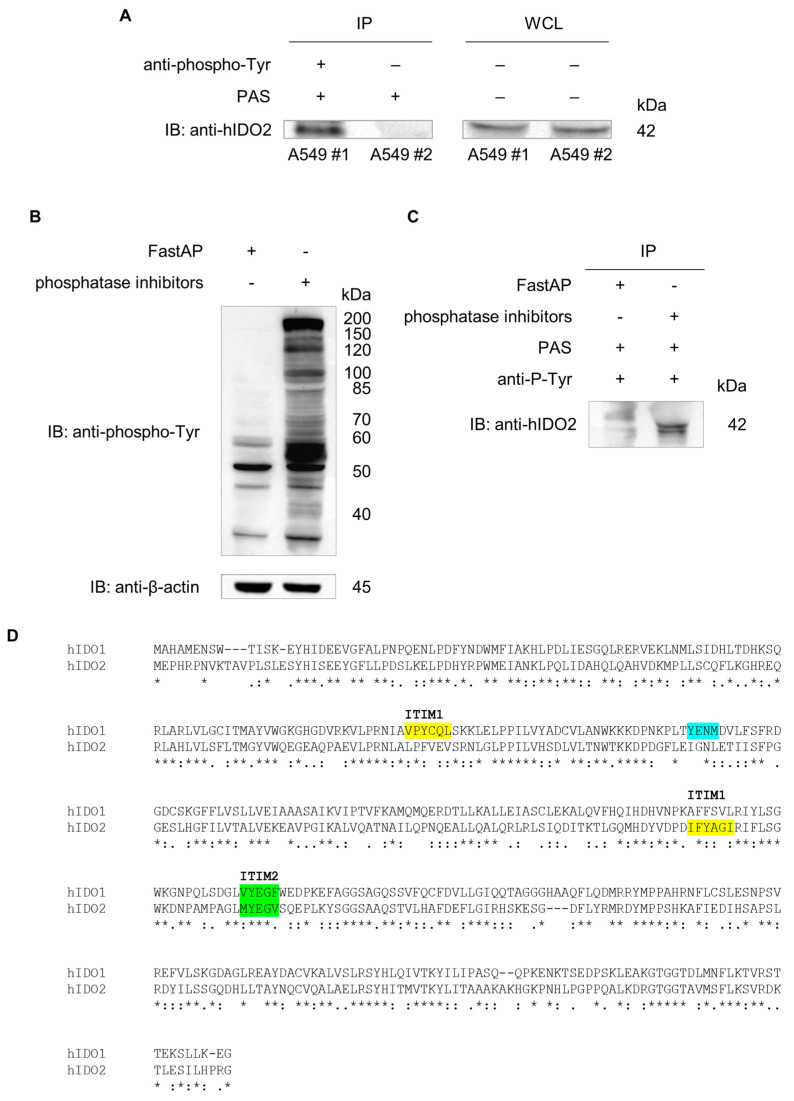
Basal phosphorylation of IDO2 tyrosine residues in A549 cells. (**A**) Whole lysate from 12 × 10^6^ untreated A549 cells was immunoprecipitated (A549 #1) using anti-phospho-tyrosine mAb mix (2 µL in 400 µL of cell lysate, overnight at 4 °C) and 30 µL of Protein-A Sepharose beads (PAS) (1 h at 4 °C under rotary agitation). The negative control (A549 #2) was prepared via incubation of the whole cell lysate with only PAS, avoiding pre-incubation with the anti-phospho-tyrosine mAb mix. Washed immunoprecipitated samples were run on SDS-PAGE and analyzed via Western blotting. Tyrosine-phosphorylated IDO2 detection was achieved using the rabbit polyclonal anti-hIDO2 antibody I23O2 (left panel). Whole cell lysates (WCL), saved before immunoprecipitation, were used as a pre-immunoprecipitation protein content control in the anti-hIDO2 immunoblot (right panel). One experiment representative of three. (**B**) A549 cell lysate was dephosphorylated or left phosphorylated by adding FastAP Thermosensitive Alkaline Phosphatase or phosphatase inhibitors, respectively. For Western blotting analysis, 10 µL was loaded with anti-phospho-tyrosine mAb mix. Β-actin was used as the loading control. (**C**) The whole cell lysate from A549 cells was split into two samples to be pre-treated with FastAP or left untreated. Then, samples were immunoprecipitated with anti-phospho-tyrosine mAb mix in presence of PAS, as in (**A**). Washed immunoprecipitated samples were run on SDS-PAGE and analyzed by Western blotting. Tyrosine-phosphorylated IDO2 detection was achieved by the rabbit polyclonal anti-hIDO2 antibody I23O2. (**D**) Human IDO1 and IDO2 amino acid sequences containing putative phosphorylation sites. ITIM1 (yellow) and ITIM2 (green) motifs are indicated in both IDO1 (accession number: NP_002155.1) and IDO2 (accession number: NP_919270.3) sequences aligned via T-coffee (https://tcoffee.crg.eu/apps/tcoffee/index.html; URL accessed on 28 February 2023). Phosphoinositide 3-kinase (PI3K) binding domain (YENM, cyan) is indicated in IDO1 protein. *, positions that have a single and fully conserved residue; conservation between groups of strongly similar properties with a score greater than 0.5 on the PAM 250 matrix; period conservation between groups of weakly similar properties with a score less than or equal to 0.5 on the PAM 250 matrix.

## Data Availability

The data presented in this study are available upon request from the corresponding author without undue reservation.

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
