# Peer review of "Membrane Localization and Phosphorylation of Indoleamine 2,3-Dioxygenase 2 (IDO2) in A549 Human Lung Adenocarcinoma Cells: First Steps in Exploring Its Signaling Function"

_ijms, 2023, doi:10.3390/ijms242216236_

Round 1

Reviewer 1 Report

Comments and Suggestions for Authors

This manuscript titled “Membrane localization and phosphorylation of indoleamine 2,3-dioxygenase 2 (IDO2) in A549 human lung adenocarcinoma cells: first steps in exploring its signaling function”, explores the membrane localization and phosphorylation of indoleamine 2,3-dioxygenase 2 (IDO2) in A549 human lung adenocarcinoma cells, aiming to shed light on its potential signaling function. The study provides interesting insights into IDO2 biology, particularly its non-enzymatic role in cancer cells.

Below concerns should be address in a revised manuscript before acceptance for publication.

  1. Figure 2A, the Immunocytochemical results should be observed by higher magnification microscopes to more detailed information. Moreover, Immunofluorescence analysis is also required for further confirmation. There is another possibility that IDO2 also located in other organelles and Immunofluorescence analysis can test this hypothesis.
  2. The experimental design for Figure 3A was inappropriate. For immunoprecipitation experiment, “Input” is important control to make sure whole cell lysates in different groups are totally equal. Moreover, a complimentary experiment is required, which is immunoprecipitating IDO2 and blotting with phospho-tyrosine antibody.
  3. The innovation of this study.

The article mentions that it aims to explore the signaling function of IDO2 but does not clearly state the research question or hypothesis. A well-defined research question or hypothesis would help guide the reader and provide a clear focus for the study. Most results of this paper are purely descriptive. More experiments and results are required for potential implications and future directions of IDO2. Further experiments (i.e. high throughout analysis, immunofluorescence, knockdown/knockout, defect phenotypic identification and pathway regulation) are required and more beneficial for interested authors.

  1. It would be beneficial to discuss the potential implications of IDO2's membrane localization and phosphorylation in A549 cells. How might these findings relate to cancer biology or immunomodulation? What are the possible downstream signaling pathways involved? Providing more context and interpretation would enhance the article's scientific value.
  2. There are some grammatical issues and awkward sentence structures throughout the article. Proofreading and editing for language and clarity would improve the overall readability of the manuscript.

Comments on the Quality of English Language

There are some grammatical issues and awkward sentence structures throughout the article. Proofreading and editing for language and clarity would improve the overall readability of the manuscript.

Author Response

Please find attached our point-by-point response to Reviewer 1's comments.

Reviewer 2 Report

Comments and Suggestions for Authors

The manuscript by Suvieri, et al. attempts at addressing the role of IDO2 protein in lung adenocarcinoma cell line A549. While IDO2 is indeed a relatively poorly characterized target and the efforts to explore it are appreciated, this study is limited with a small range of performed experiments and does not represent a significant contribution to the field.

Major issues:

1. Judging by the images of full membranes, the anti-IDO2 antibody has poor selectivity. Several bands are coloured at a wide range of molecular weights and the authors choose one of the bands just based on the approximate Mw of the protein of interest. The authors should have used a more selective antibody or at least demonstrated the absence of antibody signal in case of protein knockdown or knockout. Gene silencing (in melanoma context) has been reported before (https://www.ncbi.nlm.nih.gov/pmc/articles/PMC5078016/), so information on the siRNA is readily available. Also, IDO2-knockout A549 cell line is commercially available: https://www.creative-biogene.com/Human-IDO2-Knockout-Cell-Line-A549-CSC-RT2187-1409608-207.html .

2. The authors do not mention the number of independent experiments in case of Fig 2. In case of other Figures, three independent experiments are mentioned – as a reviewer, I would like to see the images of all such experiments in the Supplementary, as for me, the current evidence is inconclusive.

3. The quality of the supplementary file is below any good publishing standards. The lines are not marked, the ladder Mw range is not marked, no legends are provided.

4. The expression of IDO2 in A549 cells has been reported before (https://journals.plos.org/plosone/article/figures?id=10.1371/journal.pone.0126159 ), so it is not a novel finding. The method chosen for assessment of protein intracellular localization is not adequate, the authors should have carried out immunocytochemistry with fluorescent antibodies and imaged the cells with much higher resolution. Currently, the reader can only see increase of signal, whereas the assessment of intracellular localization based on the current images is physically impossible.

5. The selectivity of the phospho-tyrosine antibody used should also be demonstrated (e.g., by increasing the signal following treatment of cells with the sodium orthovanadate or other inhibitor of phospho-Tyr phosphatases). Currently, a faint band might just indicate the non-specific binding of an antibody. Also, the authors should carry out point-mutations in the IDO2 construct to identify the Tyr residue(s) phosphorylated. As of now, the information provided gives too few input for other researchers to rely on.

6. The authors should use more than one NSCLC cell line to even start drawing conclusions on the putative importance of IDO2 in NSCLC.

Author Response

Please find attached our point-by-point response to Reviewer 2's comments.

Round 2

Reviewer 1 Report

Comments and Suggestions for Authors

The authors made improvements in this version, including additional data. No further question or concern for this revised manuscript.

Author Response

We thank Reviewer 1 for the suggestions and for allowing an improvement to the manuscript.

Reviewer 2 Report

Comments and Suggestions for Authors

I appreciate the effort made by the authors to answer my previous questions, but I think that the authors should incorporate the extra images provided to Reviewers only (the siRNA experiment, the phosphatase experiment, and the fluorescence microscopy data) into the main text or Supplementary part of the manuscript. The fact that the manuscript is a brief report does not mean that its scientific soundness should remain questionable for the readers.

As to the fluorescence microscopy images, please also provide images from the bright-field channel or utilize a plasma membrane marker as a reference. I have worked with A549 cells for a long time and from the current images, I would say that the hIDO2 signal is NOT located in/close to the plasma membrane, but rather in the area surrounding the nucleus – maybe the endoplasmic reticulum or Golgi. The interaction with E-cadherin claimed based on the sucrose gradient experiment does not show the plasma membrane-localization of the protein, as E-cadherin can also be found in Golgi and ER (see e.g. https://www.tandfonline.com/doi/full/10.1080/15419060802460748 ).

Also, I consider statements such as “we found IDO2 transcript and protein expression in many human cell lines, such as HepG2, HEK293, H1975, H1650, SKOV-3, and Caco2 (data not shown)” (lines 307-308) inconclusive for the readers – please either provide data (including the Western blot, which has not been shown to reviewers, either) or remove this sentence from the text. Furthermore, I still think that a sentence regarding the fact that the authors could “find additional evidence for a possible IDO2 signaling role” is not substantiated, given the absence of follow-up studies on the Tyr phosphorylation status and the inconclusive data on the intracellular localization of the protein.

Please also declare clearly the limitations of the current study (including the poor selectivity of the used antibody) at the end of the discussion part.

Round 3

Reviewer 2 Report

Comments and Suggestions for Authors

Given the fact that antibody staining on the Western membrane yielded so many bands, the used antibody is undoubtedly non-selective. I am still asking authors to state this clearly in the manuscript. The antibody might, among other targets, recognise also protein of interest, but claiming that antibody is selective is inadequate in the present case and gives confusing information for the readers who may not have time to look into the Supplementary for the full membrane images. Alternatively, the authors could transfer one of such images with numerous extra bands to the main text so that the readers can clearly see the poor selectivity of the antibody themselves.

The statement regarding the IDO2 plasma membrane localization (lines 167-168) should also be rephrased according to the authors response to my previous review: please add "we cannot rule out the possibility of a localization in other membranes, such as the Golgi and ER membranes".
